# Effects of Different Scan Projections on the Quantitative Ultrasound-Based Evaluation of Hepatic Steatosis

**DOI:** 10.3390/healthcare10020374

**Published:** 2022-02-14

**Authors:** Laura De Rosa, Antonio Salvati, Ferruccio Bonino, Maurizia Rossana Brunetto, Francesco Faita

**Affiliations:** 1Institute of Clinical Physiology, National Research Council, 56124 Pisa, Italy; laura.derosa.95@gmail.com; 2Department of Information Engineering and Computer Science, University of Trento, 38122 Trento, Italy; 3Hepatology Unit and Laboratory of Molecular Genetics and Pathology of Hepatitis Viruses, Reference Centre of the Tuscany Region for Chronic Liver Disease and Cancer, University Hospital of Pisa, 56124 Pisa, Italy; salvatiantonio@hotmail.com (A.S.); maurizia.brunetto@unipi.it (M.R.B.); 4Institute of Biostructures and Bioimaging, National Research Council, 80145 Naples, Italy; ferruccio.bonino@unipi.it

**Keywords:** ultrasound, hepatic steatosis, NAFLD, hepatic–renal ratio, steato-score

## Abstract

Non-alcoholic fatty liver disease (NAFLD) is becoming a global public health issue and the identification of the steatosis severity is very important for the patients’ health. Ultrasound (US) images of 214 patients were acquired in two different scan views (subcostal and intercostal). A classification of the level of steatosis was made by a qualitative evaluation of the liver ultrasound images. Furthermore, an US image processing algorithm provided quantitative parameters (hepatic–renal ratio (HR) and Steato-score) designed to quantifying the fatty liver content. The aim of the study is to evaluate the differences in the assessment of hepatic steatosis acquiring and processing different US scan views. No significant differences were obtained calculating the HR and the Steato-score parameters, not even with the classification of patients on the basis of body mass index (BMI) and of different classes of steatosis severity. Significant differences between the two parameters were found only for patients with absence or mild level of steatosis. These results show that the two different scan projections do not greatly affect HR and the Steato-score assessment. Accordingly, the US-based steatosis assessment is independent from the view of the acquisitions, thus making the subcostal and intercostal scans interchangeable, especially for patients with moderate and severe steatosis.

## 1. Introduction

The prevalence of non-alcoholic fatty liver disease (NAFLD) is reaching epidemic proportions worldwide: in Western countries, NAFLD affects 20–30% of the overall population [1,2] and is also highly prevalent among children and adolescents [3]. NAFLD includes a wide spectrum of pathologies starting from simple steatosis that can be exacerbated by inflammation, thus leading to non-alcoholic steato-hepatitis (NASH), which in turn can further progress to fibrosis, cirrhosis and hepatocellular carcinoma [4,5].

The evaluation of NAFLD is of great importance not only as concerns the hepatic function, but also to obtain information about the global health condition of patients, especially from a cardio-metabolic point of view. Indeed, NAFLD is considered the hepatic manifestation of the metabolic syndrome being strongly associated with obesity, diabetes, insulin resistance, dyslipidemia and hypertension [6,7,8]. Furthermore, previous studies have pointed out that NAFLD represents a risk factor for cardiovascular disease [9], as it is also associated with subclinical markers of vascular damage [10,11,12]. Beyond its relationship with cardiovascular disease, previous studies have also underlined as NAFLD is linked to multiple risk factors for chronic kidney disease [13].

In view of all the previously mentioned associations, a simple and easily available tool for the assessment of hepatic steatosis severity becomes of great importance. The gold standard technique for the evaluation of fatty liver is liver biopsy [14]; however, this approach is invasive, suffers from sampling problems and is influenced by the subjectivity and experience of the pathologist [15]. Alternatively, different non-invasive imaging methods have been proposed for the quantification of the hepatic fat content [16]. Proton magnetic resonance spectroscopy (^1^H-MRS) represents a valid approach for estimating steatosis in a non-invasive, non-ionizing and accurate way [17], and the correlation with biopsy measurements have been already demonstrated [18]. However, this technique is characterized by high costs and low availability, thus being not suitable for employment on large cohorts of high-risk patients or in screening and follow-up programs.

Ultrasound (US) is a non-invasive, relatively inexpensive and widely available imaging technique that can be used for the assessment of hepatic steatosis [19]: in fact, the presence of fat in the liver causes an increase in US echogenicity, as well as a reduction in the penetration of the US beam, which results in an impaired visualization of deep structures, such as the diaphragm line [20]. In clinical practice, these features are usually evaluated in a qualitative way [21], leading to an operator-dependent assessment. In order to overcome this limitation, quantitative US methods have been proposed. These approaches involve the quantitative evaluation of imaging parameters by processing US images. Among these, one of the most used US parameters is the hepatic–renal ratio (HR), which represents the ratio between the echogenicity of the renal cortex and the liver parenchyma. This approach is based on the assumption that an increased fat content intensifies the echogenicity of the liver leaving the echogenicity of the renal cortex unchanged [22]. This parameter has been employed alone or in combination with other US parameters [23,24] to estimate the hepatic fat content. More recently, a new US-based score, e. g. the Steato-score, has been proposed and validated. Steato-score combines HR value with other three US-derived parameters (Attenuation Rate, Diaphragm Visualization and Portal Vein Wall Visualization, which represent the measurement of the US beam attenuation into the liver parenchyma, the level of sharpness of the diaphragm line and the contrast between portal view wall/liver parenchyma, respectively) [25].

HR can be evaluated by processing US images acquired both from the right longitudinal subcostal and the right longitudinal intercostal view [26]. However, these two projections are characterized by distinctive relative positions of both the liver and the kidney, thus presenting different paths of the US beam reaching the target organ, especially in the case of high body mass index (BMI) values and/or in presence of intestinal gas. These differences can potentially influence the assessment of HR, as well as the evaluation of more complex and informative indexes based on the combination of HR with other parameters, such as the Steato-score.

The aim of this study is to investigate the effects of the two different scan projections (i.e., the right longitudinal subcostal and the right longitudinal intercostal) on the quantitative US-based assessment of the hepatic steatosis. In particular, the influence of the scan projection was tested in terms of evaluation of the HR alone or in combination with other US parameters (Steato-score).

## 2. Materials and Methods

### 2.1. Study Population

The study population included 214 subjects enrolled at the University Hospital of Pisa who gave their informed consent; the study protocol was approved by the Ethic Committee of the University Hospital of Pisa (19 February 2016). All scans and measurements were performed by a single skilled operator in a temperature-controlled room (22–24 °C).

### 2.2. US Examination

US scans were obtained for all the participants involved in the study using a standard US system (LogiQ E9, GE Healthcare, Buckinghamshire, UK) equipped with a 1.8–5 MHz convex probe. US images were recorded in breath-hold with the subject in supine/lateral position using three different scan projections: the right longitudinal subcostal view, the right oblique subcostal view and the right longitudinal intercostal view. In each case, two consecutive 5 s long clips were stored. All spatial and temporal filters were disabled and the time gain compensation (TGC) control was maintained neutral for all scans in order to acquire raw US B-mode images. The overall gain, as well as the image depth were optimized for each participant since the assessment of the US parameters accounts for possible differences in this acquisition parameters.

### 2.3. US Parameters Calculation

In order to test the influence of the scan projection on the HR evaluation, both considering it as a single parameter or in combination with other US imaging biomarkers for a US-based intrahepatic fat content assessment, the US images were post-processed by a customized software developed using Matlab R2016b (MathWorks Inc., Natick, MA, USA).

In particular, HR was assessed from both the longitudinal subcostal and the longitudinal intercostal view (HR_sub_ and HR_inter_, respectively). The HR_sub_ measurement was obtained as previously reported in [25]. Briefly, two regions of interest (ROIs) were manually placed on the liver parenchyma and in the renal cortex, respectively, avoiding hypo and hyperechoic focal zones as well as portions of tissue, including large vessels. The two ROIs were positioned at the same depth and as close as possible to the center of the image, in order to minimize possible effects related to the depth-dependent echo intensity attenuation and the borderline echo distortion. A horizontal line is automatically drawn by the software to guide the placement of the two ROIs at similar depth. For each frame, HR_sub_ was evaluated as the ratio between the mean grey level value of the hepatic and renal ROIs; the final HR_sub_ value was obtained averaging the measurements obtained for each single frame. Regarding HR_inter_ assessments, they were obtained following the same procedure implemented for the HR_sub_ evaluation; however, in this case, due to the different relative anatomical position of the liver and the kidney within the US image, a perfect depth alignment of the two ROIs is not achievable (Figure 1). Additionally, in this case, the software reports vertical line to align the ROIs. For both the HR_inter_ and the HR_sub_ assessments, hepatic ROIs were placed in homogeneous regions in order to avoid zones with vessels or other structures. The ROIs’ size was also variable depending on the dimensions of these homogeneous areas. In general, the only limitation of the two scan views for assessing HR parameter is linked to the maximal depth that could be scanned by the US device.

Three additional US parameters were calculated. In particular, Attenuation Rate (AR) and Diaphragm Visualization (DV) were assessed from the US clips acquired in oblique subcostal view as already described in [25]. Briefly, the attenuation of the US beam crossing the liver parenchyma, i.e., the AR, was assessed obtaining grey-level vertical profiles and fitting them with a decreasing exponential curve [23], thus estimating the decay constant. As concerns DV values, they were evaluated as the maximum of the mean profile crossing the diaphragm line normalized for the overall gain and the depth at which this line is located in the B-mode US image. Finally, the portal vein wall visualization (PVWvis) was assessed as the ratio between the mean grey level obtained from a ROI correspondent to the portal vein near wall and from a second ROI placed in the liver parenchyma. The acquisition of the US images used for the calculation of the three additional parameters (AR, DV, and PVWvis) does not imply significant changes of the scan projection, thus not requiring a specific analysis as performed for the HR evaluation.

### 2.4. Steato-Score Assessment

HR, AR, DV and PVWvis were linearly combined in order to obtain a single US score representative of the hepatic fat content. As previously reported [25], the linear combination was obtained employing a multivariate backward stepwise regression analysis using MRS-derived hepatic fat content as gold standard. In order to evaluate the effects of different HR evaluations on the multiparametric score, the Steato-score was calculated using both HR_sub_ and HR_inter_, according to the following equations:Steato-score_sub_ = 3.83 + 5.35 × HR_sub_ + 153.28 × AR − 28.24 × DV − 1.27 × PVWvis(1)
Steato-score_inter_ = 3.83 + 5.35 × HR_inter_ + 153.28 × AR − 28.24 × DV − 1.27 × PVWvis(2)

### 2.5. Qualitative US Evaluations

The same skilled operator graded each US examination on the base of the presence and severity of liver steatosis. The qualitative classification into four classes of steatosis severity was obtained in 205 of the 214 total patients. In particular, each subject was classified within the following four classes: absence of steatosis (S0), mild steatosis (S1), moderate steatosis (S2) and severe steatosis (S3). Classification criteria were based on [1,22].

### 2.6. Statistical Analysis

Data are presented as median [interquartile range]. The normal distribution of HR_sub_, HR_inter_, Steato-score_sub_ and Steato-score_inter_ was tested employing the Shapiro–Wilk test.

The correlation between HR_sub_ and HR_inter_, as well as for that between Steato-score_sub_ and Steato-score_inter_, was tested by means of the Spearman correlation coefficient. Bland–Altman analysis was also performed and the presence of possible trends was evaluated employing the Spearman correlation coefficient. Furthermore, in both the cases, Intra-Class Correlation (ICC) coefficients were evaluated.

In order to test possible dependencies from the BMI values, all the analyses were repeated splitting the population according to the BMI value (BMI < 25 kg/m^2^ and BMI ≥ 25 kg/m^2^). Similarly, all the analyses were repeated dividing the whole population in the four classes of steatosis identified by means of the qualitative US assessment: absence of steatosis, mild, moderate and severe steatosis (S0, S1, S2, S3, respectively). In addition, the subjects were divided into three groups according to the tertile values of both HR and Steato-score differences: in both cases, the BMI values of the three groups were compared employing the Kruskal–Wallis test.

All the statistical tests were considered significant for *p* < 0.05. SPSS Version 23 (IBM, New York, NY, USA) was used for all statistical analyses.

## 3. Results

Table 1 shows the main characteristics of the study population, considering all the subjects, as well as dividing them on BMI values and on qualitative US classes. HR_sub_, HR_inter_, Steato-score_sub_ and Steato-score_inter_ were non-normally distributed (*p* > 0.001 for Shapiro–Wilk test). Accordingly, non-parametric tests were performed for the subsequent analyses.

HR_sub_ and HR_inter_, as well as Steato-score_sub_ and Steato-score_inter_ were significantly correlated (R = 0.89, *p* < 0.001, and R = 0.96, *p* < 0.001, respectively, as shown in Figure 2a,c). Table 2 reports the results of the Bland–Altman analysis, while Figure 2b,d show the two Bland–Altman plots. For both the HR and the Steato-score comparisons, the biases were not significant; furthermore, no significant trend were found in both cases (R = 0.02, *p* = 0.80, and R = −0.06, *p* = 0.45, respectively). ICC values were equal to 0.88 for the HR assessment and to 0.96 for the Steato-score evaluation. 

Similar results were found splitting the population according to the BMI value: significant correlations were found for the two BMI classes for both HR and Steato-score evaluations (BMI < 25 kg/m^2^: R = 0.81, *p* < 0.001, and R = 0.95, *p* < 0.001, respectively; BMI ≥ 25 kg/m^2^: R = 0.87, *p* < 0.001, and R = 0.95, *p* < 0.001, respectively). As shown in Table 3 the biases from the Bland–Altman analyses for the two BMI classes were non-significant for both HR and Steato-score assessments (Bland–Altman plots obtained by stratifying the population in the two classes by BMI values using a cutoff equal to 25 kg/m^2^ are reported in Appendix A). For each case, no significant trends were present for both HR and Steato-score comparisons (BMI < 25 kg/m^2^: R = 0.22, *p* = 0.09, and R = 0.22, *p* = 0.09, respectively; BMI ≥ 25 kg/m^2^: R = 0.13, *p* = 0.15, and R = 0.03, *p* = 0.77, respectively). ICC values were equal to 0.84 for the HR assessment and to 0.95 for the Steato-score evaluation in the case of BMI < 25 kg/m^2^, while for subjects with BMI ≥ 25 kg/m^2^ they accounted for 0.88 and 0.94, respectively.

As concerns the partition of the study population according the qualitative US assessment classes, significant correlations were found for all the classes for both HR and Steato-score assessments (S0: R = 0.65, *p* = 0.001, and R = 0.90, *p* = 0.001, respectively; S1: R = 0.72, *p* = 0.001, and R = 0.92, *p* = 0.001, respectively; S2: R = 0.86, *p* = 0.001, and R = 0.95, *p* = 0.001, respectively; S3: R = 0.81, *p* = 0.001, and R = 0.90, *p* = 0.001, respectively). However, the Bland–Altman analysis highlights some differences between the groups. In particular, the biases were significant for both HR and Steato-score evaluations in the case of S0 and S1, while they remained non-significant in the case of S2 and S3, as reported in Table 4. For each case, no significant trends were found for both HR and Steato-score evaluations (Table 4). Concerning HR assessments, ICC values were equal to 0.75 for S0, 0.67 for S1, 0.81 for S2, and 0.84 for S3; the same analysis performed for Steato-score assessments provided values equal to 0.91 for S0, 0.93 for class 1, 0.93 for S2, and 0.91 for S3. Bland–Altman plots obtained by stratifying the whole population in the four classes obtained by the qualitative classification of steatosis level (S0-S1-S2-S3), for both HR and Steato-score are reported in the Appendix A.

## 4. Discussion

In the present work, we investigated the effects of two different US scan projections, i.e., the right longitudinal subcostal and the right longitudinal intercostal, on the quantitative US-based assessment of hepatic steatosis. In particular, we tested the influence of the two different scans on the assessment of the HR parameter alone or when used in combination with other US parameters for the assessment of the multiparametric Steato-score.

NAFLD includes a wide spectrum of pathological conditions: simple steatosis can be exacerbated by inflammation and fibrosis, thus advancing to NASH, and potentially progress further and evolve into cirrhosis and liver cancer [4,5]. Previous evidence has suggested that this pathological condition is associated with the metabolic syndrome [6,7,8] and represents a risk factor for the development of cardiovascular [9] and chronic kidney diseases [13]. For these reasons, the estimation of the hepatic fat content plays a key role not only in the assessment of the liver function, but also in the evaluation of the overall health condition, with particular reference to the cardio-metabolic risk. From a technical point of view, MRS imaging represents the non-invasive gold standard for the quantification of the hepatic fat content [17,18], but its adoption is limited by the high costs and the low availability. Highly available and relatively inexpensive US imaging can represent a valid alternative, but it is usually limited to qualitative analysis that suffers from operator dependency and low reproducibility, thus leading to partially unreliable results [27]. In order to overcome this limitation, some efforts have been made towards the assessment of quantitative US parameters representative of the steatosis grade; in particular, HR value, used as single index [22,28] or in combination with other US parameters [23,24,25], is one of the most employed. Despite the advantage of quantitative evaluation, a certain level of variability in the evaluation of HR can be introduced by the scan projection. Therefore, the standardization in terms of US views during the image acquisition phase is of great importance.

Considering the whole population, the comparisons between the subcostal and intercostal assessments provided significant correlations for both HR and Steato-score, as well as no significant biases and absence of significant trends in the Bland–Altman analysis. These results were confirmed by the ICC values, which highlighted an excellent agreement between the two measurements [29], both considering HR individually or combined with other US parameters for the calculation of the Steato-score. These results suggest that the scan projection does not affect the HR evaluation, thus making the subcostal and intercostal scans interchangeable.

From previous studies, it is known that BMI could affect the sensibility of US-based approaches in the evaluation of the hepatic steatosis grade [30,31] and that the higher thickness of subcutaneous adipose tissue, characterizing overweight and obese subjects, might introduce challenges when performing an US examination, with particular reference to the visualization of deep structures [30]. However, our data suggest that differences in the thickness of the subcutaneous adipose tissue between the US probe and the organs of interest characterizing the two tested scan projections do not have impact on both HR and Steato-score assessment. This result is also confirmed when the population is divided in different classes of HR and Steato-score values as no significant statistical differences between groups were found. This finding could be due to the fact that the quantitative estimations performed in the present study can strengthen the measurement and reduce its variability induced by differences in tissues crossed by the US beam before reaching the target organ. Indeed, it has been already shown that the Steato-score preserves its diagnostic performance even in the case of overweight and obese subjects [25]. Furthermore, the capability of quantitative US-based liver steatosis assessment in overcoming effects induced by different scan projections was also demonstrated for Acoustic Structure Quantification (ASQ) US parameter, which is based on the echo amplitude distribution [32]. In addition, overweight and obese conditions have been associated with an overall reduced US image quality [33,34] with no differently impact on the two scan projections and, therefore, not affecting our findings.

However, some differences have emerged when dividing the population on steatosis classes. Significant correlations were found for both HR and Steato-score for all the classes, but significant biases are present in the case of absent or mild steatosis. This can be due to the fact that, supposing the same mean grey level value for the renal ROI [22], the projection-induced difference on the mean grey level measurements of the hepatic ROI has a greater impact on the lower HR values (absent or mild steatosis) compared to the HR values, evaluated in cases of moderate or severe steatosis, the latest being generally higher because of a more hyperechoic parenchyma. Furthermore, it should be underlined that moderate and severe steatosis classes are associated with wider samples in our study, which strengthen the statistical power of the bias calculations. Another possible source of discrepancy between the two assessments can be due to the non-uniform distribution of the liver fat, which has been already reported [35] and might affect the two evaluations in a different way. 

General considerations need to be taken into consideration as concerns the ICC values obtained. All considered cases suggest that the measurement obtained from the two different scan projections are in excellent agreement, except for the HR evaluation in subjects without steatosis, for which the agreement is good [29]. However, it should also be noted that ICC values for the Steato-score are higher than those calculated for the HR. Furthermore, if a different reference for ICC cut-offs were to be adopted [36], the agreement between the two measurements would offer good results for the HR assessments and would remain excellent for the Steato-score evaluations. This consideration suggests that the Steato-score, being based on the combination of HR with other US parameters, is less sensitive to differences in the US scan projections. Accordingly, the introduction of a multi-parametric approach, beyond being more representative of the complete US pattern related to the hepatic fat content, can lead to a reduction in the variability of the steatosis grade assessment induced by different scan projections. Another important point is the multi-parametric nature of the Steato-score. Although the availability of multiparametric techniques is still limited, in [25] we demonstrated that the Steato-score is much more accurate in quantifying the level of steatosis compared to single parameters. This is also the reason why the effects of different scan projections are reduced when the HR parameter is embedded into a multiparametric equation together with other US parameters.

This study has some limitations. Firstly, the lack of a sub-population including obese subjects only, i.e., characterized by a BMI ≥ 30 kg/m^2^ [37], does not allow conclusions to be drawn about the influence of the scan projection on the HR and Steato-score assessments in the case of greater subcutaneous adipose tissue thickness. A second limitation concerns the partition of the population in steatosis classes, which was performed on the basis of a qualitative US assessment by a skilled and experienced operator. The non-invasive evaluation of the real hepatic fat content by means ^1^H-MRS, which is considered the non-invasive gold standard for the assessment of the steatosis grade, would have provided a more accurate and operator-independent population classification, thus allowing a more scrupulous investigation of the influence of the scan projection among the single steatosis classes. In addition, this evaluation would have supplied the real hepatic fat content for each patient, thus enabling the identification of the scan projection providing the most accurate hepatic steatosis estimation. Furthermore, the qualitative classification of the subjects led to steatosis classes that were not equally represented, with the groups characterized by lower steatosis grades that turned out to be less represented. A more balanced distribution of the participants among the steatosis classes would have provided statistically stronger results, especially as concerns the Bland–Altman analysis. Finally, phantom experiments to test the effect of the placement of ROIs at different depths were not performed.

## 5. Conclusions

In conclusion, the presented data suggest that, in the case of absence of or mild steatosis, the choice of the scan projection during the acquisition influences both the HR and the Steato-score assessments. However, a complete interchangeability of the two scan projections was found for higher hepatic fat contents, which are those cases in which a change from the standard view could be more probably required, representing conditions more often associated with higher BMI values and worse image quality. It is known that in order to obtain more accurate quantitative US-based evaluations of the hepatic fat content, the acquisition procedure needs to be standardized and the scan projection has to be kept unvaried in order to compare results from different groups and studies. However, we demonstrated that when the anatomy of the patient does not allow to acquire standard subcostal US scan view for HR assessment, an alternative intercostal scan projection could be used thus providing a reliable steatosis assessment even in these challenging patients. Furthermore, the employment of US multi-parameters approaches, such as the Steato-score, should be preferred to the single HR calculation, since they allow US-based steatosis grade evaluations that are less sensitive to differences in the scan projection.

## Figures and Tables

**Figure 1 healthcare-10-00374-f001:**
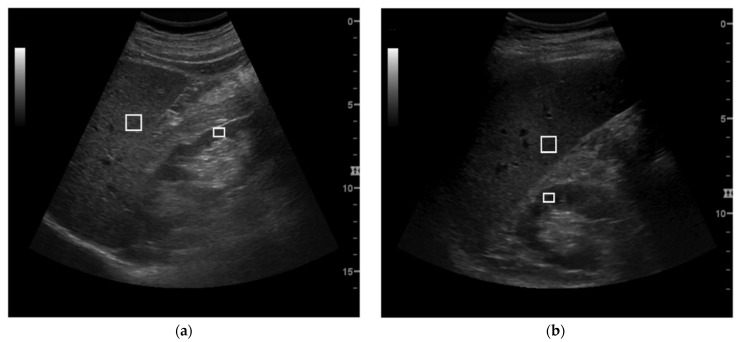
Examples of ultrasound images acquired for the hepatic–renal ratio (HR) evaluation in subcostal (**a**) and intercostal view (**b**). The two ROIs were placed at 6/6.25 cm (**a**) and 6/9 cm (**b**), respectively.

**Figure 2 healthcare-10-00374-f002:**
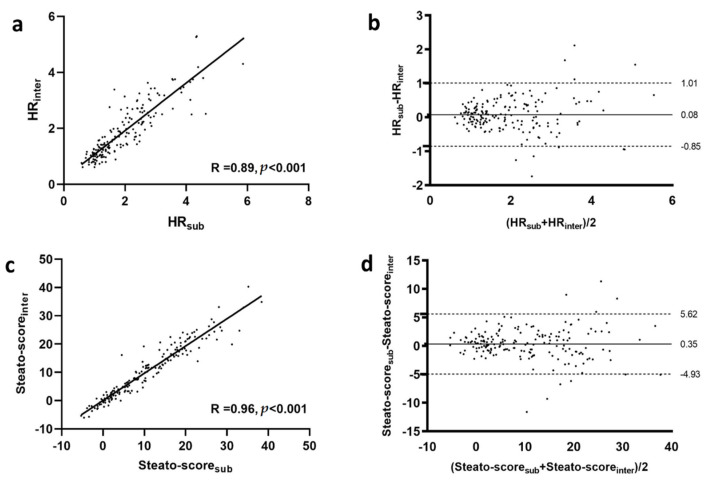
Correlation (**a**,**c**) and Bland–Altman plots (**b**,**d**) obtained for HR and Steato-score assessment, respectively. In the Bland–Altman graphs the two dashed lines represent the limits of agreements from −1.96 × SD to +1.96 × SD, and the continuous line represents the average difference in measurements (bias). HR: hepatic–renal ratio; SD: standard deviation; R: Spearman correlation coefficient.

**Table 1 healthcare-10-00374-t001:** Principal characteristics of the study population.

	Sample Size	Percentage of Males (%)	Age (Years)	BMI (kg/m^2^)
Entire population	214	55.6	53.8 ± 12.9	28.1 ± 5.5
BMI classes				
BMI < 25 kg/m^2^	65	52.3	52.9 ± 13.3	22.4 ± 1.8
BMI ≥ 25 kg/m^2^	149	57	54.3 ± 12.8	30.6 ± 4.6
Steatosis classes by qualitative US				
Absent (class 0)	42	40.5	57.1 ± 12.6	23.6 ± 2.9
Mild (class 1)	26	42.3	55.2 ± 12.6	25.4 ± 3.2
Moderate (class 2)	57	63.1	52.9 ± 13.4	27.9 ± 5.2
Severe (class 3)	80	61.2	52.0 ± 13.2	31.5 ± 5.2

Data are presented as mean ± standard deviation. BMI: body mass index.

**Table 2 healthcare-10-00374-t002:** Results of the Bland–Altman analysis for the entire population.

	Bias	SD	Upper Limit	Lower Limit
HR	0.06	0.50	1.05	−0.92
Steato-score	0.35	2.69	5.62	−4.92

SD: standard deviation of the differences; HR: hepatic–renal ratio. The limits of agreement were calculated as bias ± 1.96 standard deviation of the differences.

**Table 3 healthcare-10-00374-t003:** Results of the Bland–Altman analysis for the two BMI classes.

	BMI < 25 kg/m^2^				BMI ≥ 25 kg/m^2^			
	Bias	SD	Upper Limit	Lower Limit	Bias	SD	Upper Limit	Lower Limit
HR	0.07	0.39	0.85	−0.70	0.08	0.55	1.14	−0.93
Steato-score	0.37	2.12	4.53	−3.78	0.33	2.93	6.08	−5.41

SD: standard deviation of the differences; HR: hepatic–renal ratio. The limits of agreement were calculated as bias ± 1.96 × SD.

**Table 4 healthcare-10-00374-t004:** Results of the Bland–Altman analysis for the four steatosis classes obtained by qualitative US and correlation coefficient (R) with relative *p*-value for HR and Steato-score trends for each class.

	Bias	SD	Upper Limit	Lower Limit	R
	S0				
HR	0.12 *	0.27	0.65	−0.41	0.23 (*p* = 0.15)
Steato-score	0.64 *	1.45	3.49	−2.19	0.14 (*p* = 0.39)
	S1				
HR	0.11 *	0.28	0.67	−0.44	0.28 (*p* = 0.18)
Steato-score	0.61 *	1.51	3.59	−2.35	0.09 (*p* = 0.69)
	S2				
HR	0.05	0.55	1.13	−1.03	−0.14 (*p* = 0.32)
Steato-score	0.27	2.95	6.06	−5.51	−0.14 (*p* = 0.31)
	S3				
HR	0.05	0.62	1.16	−1.05	0.10 (*p* = 0.40)
Steato-score	0.13	3.32	6.65	−6.38	−0.28 (*p* = 0.82)

SD: standard deviation of the differences; HR: hepatic–renal ratio; S0: absence of steatosis; S1: mild steatosis; S2: moderate steatosis; S3: severe steatosis. The limits of agreement were calculated as bias ± 1.96 × SD; R: Spearman correlation coefficient. * indicates a significant bias.

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
