# Peer review of "Effects of Different Scan Projections on the Quantitative Ultrasound-Based Evaluation of Hepatic Steatosis"

_healthcare, 2022, doi:10.3390/healthcare10020374_

Round 1

Reviewer 1 Report

In this paper, the authors evaluate the differences in the classification of the level of hepatic steatosis by using different scan views (subcostal and intercostal) and showed that two different scan projections do not create a significant effect on the assessment of hepatic-renal ratio (HR) and Steato-score, however significant differences between the two parameters were observed only for patients with no steatosis or a mild level of it.

However, I have some concerns regarding the experimental steps and the novelty of the manuscript by the authors. First of all, in figure 1, what is the exact depth for these two white ROI with the unit? For the general readers, I think this information should be given in the text or at least should be mentioned into the figure description besides representing the figure. I am estimating that even if the same depth for two scan views is achievable for the region of interest, the depth for each patient should be different due to the non-uniform distribution of the liver fat. I think the depth distribution for 214 patients should be shown to see more details. Additionally, I would like to read more information on how to manage to put the same position for ROI in-depth direction. Before collecting the data from the patient, is it done any phantom experiment for adjusting the same depth for two scan views? Is there any limitation regarding depth capability for two scan views?

According to the result in the manuscript, especially the conclusion part, the authors claimed that “the acquisition procedure needs to be standardized and the scan projection has to be kept unvaried in order to compare results from different groups and studies”, so this sentence implied that the study did not complete yet and maybe we cannot conclude that both scan projection should be acquired for each patient to evaluate inaccurate way or not? I would like to see a certain conclusion instead of implying that the acquisition procedure needs to be standardized.

Author Response

In this paper, the authors evaluate the differences in the classification of the level of hepatic steatosis by using different scan views (subcostal and intercostal) and showed that two different scan projections do not create a significant effect on the assessment of hepatic-renal ratio (HR) and Steato-score, however significant differences between the two parameters were observed only for patients with no steatosis or a mild level of it.

However, I have some concerns regarding the experimental steps and the novelty of the manuscript by the authors. First of all, in figure 1, what is the exact depth for these two white ROI with the unit? For the general readers, I think this information should be given in the text or at least should be mentioned into the figure description besides representing the figure.

Authors’ response: Thank you for your comment. Accordingly, we added information about the ROIs depth in the caption of the Figure 1. In particular, the two ROIs were placed at 6-6.5 cm on the subcostal scan view, and at 6-9 cm in the intercostal scan view, for the hepatic and renal ROI respectively.

I am estimating that even if the same depth for two scan views is achievable for the region of interest, the depth for each patient should be different due to the non-uniform distribution of the liver fat. I think the depth distribution for 214 patients should be shown to see more details. Additionally, I would like to read more information on how to manage to put the same position for ROI in-depth direction.

Authors response: Thank you for your accurate analysis on this methodological point. Sure, it is most important to discuss about this. It is right to observe that the two ROIs depth differs slightly between the patients. This is probably due to the fact that the anatomical position of the kidney depends on the structural body conformation of each patient more than on the not-uniform distribution of the fat. The most important matter is, at least into standard scan view (subcostal view), to place the two ROIs as much as possible at the same depth. This is easy achievable with our Matlab software because, after placing the renal ROI, a horizontal line automatically appears to guide the sonographer in positioning the liver ROI. However, some small differences in the depth are allowed as another important point is to draw ROIs covering a homogeneous parenchyma without small vessels and other structures. This concept was added to the manuscript.

On the other hand, the same procedure cannot be followed with intercostal scan views, for which we allow sonographer to draw the ROI with a vertical alignment. Accordingly, one of the aims of the paper is studying the difference in assessing the hepatic-renal ratio when a different scan (i. e. intercostal) from the standard subcostal view is used.

As regards depth value distribution, this information has not been recorded during experimental analysis. However, we performed a new analysis in a subgroup of subjects (N=20) obtaining a mean value of 6.85 cm (range 6.4-7.8 cm).

Before collecting the data from the patient, is it done any phantom experiment for adjusting the same depth for two scan views?

Authors’ response: Thank you for the stimulation proposal. No, we didn’t and we are planning to do in the future. At the moment, we added this point as a limitation of the study (L333-L335).

Is there any limitation regarding depth capability for two scan views?

Authors’ response: No, there are no specific limitations for the two scan views. There is only a limitation linked to the anatomical/morphological patient conformation, and consequently to the maximum depth that the US device is able to provide. However, in this study cohort we never had this kind of limitation; although the BMI reaches values up to 47 kg/m2. We added this information in the Materials and Methods section (L128-L129).

According to the result in the manuscript, especially the conclusion part, the authors claimed that “the acquisition procedure needs to be standardized and the scan projection has to be kept unvaried in order to compare results from different groups and studies”, so this sentence implied that the study did not complete yet and maybe we cannot conclude that both scan projection should be acquired for each patient to evaluate inaccurate way or not? I would like to see a certain conclusion instead of implying that the acquisition procedure needs to be standardized.

Authors response: With the expression “the acquisition procedure needs to be standardized and the scan projection has to be kept unvaried in order to compare results from different groups and studies” we referred to the fact that, in ideal cases, all scans for steatosis assessment should be acquired following a standardized protocol (as the one we already proposed in Di Lascio et al. (ref [25] in the manuscript). However, in some real cases (especially for the patients who have a higher level of fat in the liver or for patients with intestinal gas at the moment of US scan) it could be very challenging obtaining a subcostal view of the kidney while an intercostal approach could be successfully managed. Our aim was to demonstrate that also in these subjects a reliable value of liver steatosis can be obtained avoiding to classify the patient as a failed measurement. This concept was added to the revised Conclusions section (L345-L348).

Reviewer 2 Report

All in all, in my opinion the article is good and may be published,only small issue need to solve. 

Fig 1. the points is big.Suggestions to narrow.

Author Response

All in all, in my opinion the article is good and may be published,only small issue need to solve.

Authors response: We are very glad and we hope to be able to satisfy all the reviews and publish this paper.

Fig 1. the points is big. Suggestions to narrow.

Authors response: Thank you for the suggestion. We thought your comment was referring to figure 2, as we found a typo in the main text of the original manuscript. We did it and actually the figure 2 appears more clear.

Reviewer 3 Report

De Rosa et al evaluated the use of two different scan projections (subcostal and intercostal) in algorithms that provide quantitative parameters for fatty liver quantification. The main finding of the paper is that these two scan projections are interchangeable for patients with moderate and severe steatosis whereas the choice of a particular scan projection influences assessment in absence of fat or mild steatosis.

This is a well written manuscript (sufficient background, adequate explanation of methods, extensive discussion of findings) but needs language editing. The methodology is sound and the presentation of the results is clear.

Comments:

  1. The abstract needs editing to improve clarity.
  2. L71 Please mention these other parameters.
  3. The population sample size is 214 but the sum of the class 0+1+2+3 samples (table 1) is 205. What is the explanation for this difference?
  4. Results section: Please replace figure 1 with figure 2 in Bland-Altman plots legend.
  5. Figure 2: Please mention in the figure legend what is depicted by the lines in the Bland-Altman plots.
  6. Results illustrated in Tables 3 and 4 could also be depicted in figures presenting the Bland-Altman plots.
  7. L235 Reference is missing.
  8. L331-334 This is an interesting point. You may transfer this comment to the discussion section and analyze it further.

Author Response

De Rosa et al evaluated the use of two different scan projections (subcostal and intercostal) in algorithms that provide quantitative parameters for fatty liver quantification. The main finding of the paper is that these two scan projections are interchangeable for patients with moderate and severe steatosis whereas the choice of a particular scan projection influences assessment in absence of fat or mild steatosis.

This is a well written manuscript (sufficient background, adequate explanation of methods, extensive discussion of findings) but needs language editing. The methodology is sound and the presentation of the results is clear.

Authors response: Thank you for your positive comments. The manuscript has been proofread thoroughly, and we sincerely hope it will meet with your approval.

The abstract needs editing to improve clarity.

Authors response: Thank you for the advice. We reviewed the abstract hoping it could be clearer in its present form.

L71 Please mention these other parameters.

Authors response: The description of the other three parameters is included in the revised manuscript (L70-L73).

The population sample size is 214 but the sum of the class 0+1+2+3 samples (table 1) is 205. What is the explanation for this difference?

Authors response: The difference is due to the fact that the qualitative evaluation by the sonographer is missing in 9 patients (of the 214 enrolled ones).

This was not specified in the main text; therefore, we added this explanation into the revised Methods Section (L153-L154).

Results section: Please replace figure 1 with figure 2 in Bland-Altman plots legend.

Authors response: Thank you, we did it. There was a typo.

Figure 2: Please mention in the figure legend what is depicted by the lines in the Bland-Altman plots.

Authors response: Thank you for your suggestion. Descriptions of the meaning of the different lines in the Bland-Altman plots have been added to the Figure 2 caption.

Results illustrated in Tables 3 and 4 could also be depicted in figures presenting the Bland-Altman plots.

Authors response: Our original choice of using table instead of graphical representation was aimed at avoiding an excessive number of graphs into the manuscript. According to the reviewer’s suggestion, we added the Bland-Altman plots to the Supplementary Materials, referring them into the main text (L198-L199 and L221-L224 for Table 3 and Table 4, respectively).

L235 Reference is missing.

Authors response: Thank you, there was a mistake. We added the bracket for the reference.

L331-334 This is an interesting point. You may transfer this comment to the discussion section and analyze it further.

Authors response: Thank you for your comment. We also think it is a crucial point. The advantage of using multiparametric rather than monoparametric models was largely discussed in DiLascio et al., ref [25] in the main text. For this reason, we decided to not stress this point too much in this paper. However, we added some sentences into Discussion to better underline this important concept (L311-316)

Round 2

Reviewer 1 Report

I thank to authors for replying to all comments made by me and the second reviewer.